# Year-Round Cultivation of *Tetraselmis* sp. for Essential Lipid Production in a Semi-Open Raceway System

**DOI:** 10.3390/md19060314

**Published:** 2021-05-28

**Authors:** Won-Kyu Lee, Yong-Kyun Ryu, Woon-Yong Choi, Taeho Kim, Areumi Park, Yeon-Ji Lee, Younsik Jeong, Choul-Gyun Lee, Do-Hyung Kang

**Affiliations:** 1Jeju Marine Research Center, Korea Institute of Ocean Science and Technology (KIOST), Jeju Special Self-Governing Province 63349, Korea; wonkyulee@kiost.ac.kr (W.-K.L.); ykyou0111@kiost.ac.kr (Y.-K.R.); cwy@kiost.ac.kr (W.-Y.C.); kt1024@kiost.ac.kr (T.K.); areumi1001@kiost.ac.kr (A.P.); leeyj0409@kiost.ac.kr (Y.-J.L.); jys23@kiost.ac.kr (Y.J.); 2Department of Ocean Science (Oceanography), KIOST School, University of Science and Technology (UST), Daejeon 34113, Korea; 3Department of Biological Engineering, Inha University, Incheon 22212, Korea; leecg@inha.ac.kr

**Keywords:** microalgae, *Tetraselmis* sp., open raceway pond, building information modeling, Korea, year-round production, areal productivity, lipid content, polyunsaturated fatty acid

## Abstract

There is increasing demand for essential fatty acids (EFAs) from non-fish sources such as microalgae, which are considered a renewable and sustainable biomass. The open raceway system (ORS) is an affordable system for microalgae biomass cultivation for industrial applications. However, seasonal variations in weather can affect biomass productivity and the quality of microalgal biomass. The aim of this study was to determine the feasibility of year-round *Tetraselmis* sp. cultivation in a semi-ORS in Korea for biomass and bioactive lipid production. To maximize biomass productivity of *Tetraselmis* sp., f medium was selected because it resulted in a significantly higher biomass productivity (1.64 ± 0.03 g/L) and lower omega-6/omega-3 ratio (0.52/1) under laboratory conditions than f/2 medium (0.70/1). Then, we used climatic data-based building information modeling technology to construct a pilot plant of six semi-ORSs for controlling culture conditions, each with a culture volume of 40,000 L. Over 1 year, there were no significant variations in monthly biomass productivity, fatty acid composition, or the omega-6/omega-3 ratio; however, the lipid content correlated significantly with photosynthetic photon flux density. During year-round cultivation from November 2014 to October 2017, areal productivity was gradually increased by increasing medium salinity and injecting CO_2_ gas into the culture medium. Productivity peaked at 44.01 g/m^2^/d in October 2017. Throughout the trials, there were no significant differences in average lipid content, which was 14.88 ± 1.26%, 14.73 ± 2.44%, 12.81 ± 2.82%, and 13.63 ± 3.42% in 2014, 2015, 2016, and 2017, respectively. Our results demonstrated that high biomass productivity and constant lipid content can be sustainably maintained under Korean climate conditions.

## 1. Introduction

Microalgae have received increasing interest as a biofuel resource in response to the rising energy crisis, depletion of fossil fuels, and climate change [1,2]. Recently, microorganisms have been proposed as a feedstock for the production of bioactive compounds, including β-carotene, phycobilin pigments, fatty acids (FAs), and amino acids [3,4,5,6,7]. Microalgae can be used in the bio-refinery process to produce high-value products that can increase the opportunities and possibilities for various industries [8].

*Tetraselmis* (Chlorophyta), which has previously been reported to contain a substantial lipid fraction, is a potentially promising bioactive feedstock, like other microalgal species [9,10]. Polyunsaturated fatty acids (PUFAs), such as omega-3 (ω-3) and omega-6 (ω-6), can be used to prevent cardiovascular disease, diabetes mellitus, rheumatoid arthritis, cognitive impairment, and inflammation [11,12,13,14,15]. In particular, alpha-linolenic acid (18:3n3, ALA) and linoleic acid (18:2n6, LA) are essential fatty acids (EFAs), as they cannot be synthesized in mammalian cells [16,17]. Humans must acquire PUFAs through their diet [17]. Increasing consumer awareness of the nutritional importance of these FAs has led to a significant increase in the global demand for omega-3 and omega-6 FAs [18]. Furthermore, the omega-6/omega-3 balance of FAs has been emphasized [19]. The ratio of omega-6 to omega-3 (ω-6/ω-3) FAs in western diets has increased to 15/1–16.7/1, indicating a reduction in the intake of omega-3 and unbalanced supplementation of FAs. This can cause cardiovascular disease, cancer, and inflammation in humans [19]. FA imbalances have been caused by the technological development of the vegetable oil industry and modern agriculture with a focus on grain feed rich in omega-6 FAs. This has led to an increase in research and demand for omega-6 FAs in livestock feed, which has increased the ω-6/ω-3 ratio even in milk and eggs [20,21]. This also includes a demand for aquaculture fish oil [19], which will soon exceed current aquaculture and fishery production [22].

Large-scale microalgae culture systems are generally classified as either closed systems, which are wholly enclosed within the vessel (photobioreactor, PBR), or open systems, which are directly exposed to the outdoor environment [23,24]. Microalgal biomass is mostly produced by PBRs or open raceway systems (ORSs) for mass production. The advantages of PBRs include the applicability of various structural designs, convenient control of culture conditions, prevention of biological contamination, and high biomass productivity. However, they require high operating costs, capital investment costs, and device maintenance, and it can be difficult to expand facilities [25,26]. In contrast, ORS cultivation has the advantages of using solar energy and CO_2_ from the atmosphere, as well as inexpensive materials for construction [27]. Open pond culture systems are usually applied as the main systems for commercial mass cultivation because these systems are more economical than closed systems. They can be broadly classified into several types, with raceway ponds being the most widely used culture systems and the cheapest to construct and operate [28].

Unfortunately, there are few commercially feasible ORSs for biomass production in Korea [29]. For industrial applications such as biodiesel, health food, animal feed, and fertilizers, it is not only important to have high algal biomass productivity but also to maintain year-round biochemical compositions [2,30]. However, the Korean climate has high seasonal variability, with high temperatures and rainfall in the summer and low temperatures in autumn and winter, limiting year-round microalgae cultivation. Thus, commercial cultivation of microalgae with ORSs in Korea requires the control of water temperature, selection of suitable microalgae species and culture medium, elevation of biomass productivity, maintenance of biochemical compositions, and technology for preventing contamination [24,29,31,32]. In 2011, we established ORSs in which open raceway ponds were installed in a greenhouse (so called semi-ORSs). The ORSs were based on environmental analysis of selected culture sites using the building information modeling (BIM) technology, introduced by the National Institute of Standards and Technology (NIST) of the US Department of Commerce in 2007 [29], to maximize biomass productivity. BIM is advantageous for reducing installation and operating costs and predicting the future environment by analyzing data using a 3D modeling method with big data and information [33,34,35]. Therefore, the application of BIM technology is considered to be beneficial for the construction of microalgae production facilities. 

In our previous study, we constructed a microalgal culture facility with ORSs inside a glass greenhouse, and *Arthrospira* (*Spirulina*) *maxima* was mass cultivated [29]. In this study, a new marine microalgal strain was used to scale-up culture systems at a different site near a thermal power plant. The aim of this study was to report the feasibility of biomass and bioactive lipid production from *Tetraselmis* sp. in semi-ORSs in Korea over a 3-year period.

## 2. Results

### 2.1. Selection of Culture Medium Concentration for High Biomass and FA Productivity

Laboratory experiments were carried out for 28 days to determine the optimal culture medium concentration. Double and triple concentrations of f/2 medium (f and f*1.5 medium) led to an increase in the final biomass production of Tetraselmis sp. The highest biomass concentration was 1.64 ± 0.03 g/L in f medium on the final day, followed by 1.47 ± 0.07 and 1.24 ± 0.05 g/L in f*1.5 and f/2 media, respectively. As shown in Figure 1a, there was a significant difference in biomass concentrations from day 10 to the final day. Compositions of FAs (%, per total FAs), and in particular omega-3, were concentration-dependent and increased from 17.33 ± 0.98 (f/2 medium) to 24.72 ± 1.80 and 33.10 ± 1.59% in f and f*1.5 media respectively, whereas saturated fatty acids (SAFAs) and mono-unsaturated fatty acids (MUFAs) significantly decreased (*p* < 0.05). Figure 1b shows the composition of the ω-3 and ω-6 FAs. The proportion of omega-3 among total FAs increased in a concentration-dependent manner in the three medium concentrations. ALA (C18:3n3) increased from 13.70 ± 0.82% (f/2 medium) to 22.23 ± 1.43% (f medium) and 27.87 ± 1.37% (f*1.5 medium), whereas eicosapentaenoic acid (C20:5n3, EPA) increased from 2.57 ± 0.15% (f/2 medium) to 4.17 ± 0.32% (f medium) and 5.23 ± 0.42% (f*1.5 medium). Regarding omega-6 compositions, there was no significant difference in LA (C18:2n6), with the composition varying from 10.83 ± 0.93% to 13.00 ± 2.35% in the three different medium concentrations. Gamma-linolenic acid (C18:3n6, GLA) significantly increased to 1.03 ± 0.25% in f*1.5 medium compared to levels in f/2 medium (0.40 ± 0.10%), but there was no significant difference between f medium and f/2 medium. Ratios of total omega-6 to total omega-3 (ω-6/ω-3) FAs decreased in a concentration-dependent manner from 0.70/1 (f/2 medium) to 0.52/1 and 0.41/1 in f and f*1.5 media, respectively. Based on its highest biomass concentration and increased omega-3 composition, f medium was selected for the large-scale cultivation of Tetraselmis sp.

### 2.2. Production of Biomass, Lipids, and FAs under Four Different Culture Scales

To investigate changes in biomass productivity, lipid content, and FA composition when scaling up the cultivation of *Tetraselmis* sp., culture experiments were carried out at different scales (i.e., four different culture volumes; Figure 2). 

There was a significant difference in biomass concentration between the culture volumes from day 16 to day 28 (*p* < 0.05). The final biomass concentration gradually decreased with culture volume (Figure 3a), with the highest concentration (1.27 ± 0.02 g/L) in the 2 L culture, followed by 1.08 ± 0.03, 0.96 ± 0.04, and 0.90 ± 0.08 g/L in the 5 L, 200 L and 40,000 L cultures, respectively. There was no significant effect of culture volume on lipid content of *Tetraselmis* sp. (*p* > 0.05), with contents of 19.3 ± 0.8%, 17.0 ± 1.2%, 19.2 ± 1.5%, and 16.8 ± 2.5% in 2 L, 5 L, 200 L, and 40,000 L cultures, respectively. FA compositions differed among culture volumes; however, there was a correlation between culture volume and FA composition (Figure 3b, *p* > 0.05). Omega-3 FA, omega-6 FA, MUFA, and SAFA made up 13.79−18.68%, 11.90−23.53%, 29.62−37.89%, and 30.21−36.48% of the total FAs in the four culture volumes, respectively. 

Figure 4 shows the compositions of ω-3 and ω-6 FAs in *Tetraselmis* sp. cells cultured in four different volumes. Compared with that in the control (2 L culture), the composition of LA significantly increased (14.85 ± 0.07%) in the 40,000 L culture, but there were no differences in ALA, EPA, and GLA (1.47 ± 0.16%, 16.6 ± 0.28%, and 1.15 ± 0.07%, respectively). The ratios of omega-6 to omega-3 (ω-6/ω-3) FAs were 0.73/1, 0.87/1, 1.37/1, and 1/1 in the 2 L, 5 L, 200 L, and 40,000 L cultures, respectively. Although there was no correlation between FA content and culture volume, the FA content was maintained in the 40,000 L volume. These results indicate that even though biomass productivity of *Tetraselmis* sp. was significantly reduced when the culture volume was scaled up, sustainable lipid production and maintenance of FA composition were feasible.

### 2.3. Culture Environment and Biomass and Lipid Productivities of Tetraselmis sp. in the ORS over 1 Year

Figure 5 shows the facility for the sustainable annual cultivation on a pilot scale, in which environmental conditions were controlled using a semi-ORS. Cultivation was carried out for 1 year to investigate the monthly areal productivity, lipid content, and FA composition of *Tetraselmis* sp. Figure 6 shows the pond (medium) temperature and photosynthetic photon flux density (PPFD) from 1 January to 31 December in the plant. The pond temperature of the ORS was controlled within a range that did not affect microalgal growth relative to large variations in air temperature. The average pond temperature was controlled at 27.35 ± 2.76 °C, and PPFD, which could not be controlled, was 189.21 ± 200.14 µmol photon m^−2^s^−1^ (with a peak of up to 911.7 µmol photon m^−2^s^−1^ in June).

Figure 7 shows monthly average areal productivity (AP) and lipid content over 1 year. Monthly AP varied from 8.30 ± 2.01 to 12.96 ± 2.65 g/m^2^/d with an annual average of 9.65 ± 2.40 g/m^2^/d. Productivity peaked in February and September. However there were no significant differences in monthly AP, and AP did not correlate with water temperature or PPFD (*p* > 0.05). Lipid content of cells was lowest in November (9.9%) and highest in May (18.2%). Lipid content was positively correlated with PPFD over the year (*p* = 0.019).

Fatty acid composition was divided into SAFA, MUFA, PUFA ω-3, and PUFA ω-6 (Figure 8). Polyunsaturated fatty acid ω-3 varied from 15.84% to 39.92%, peaking in March, whereas ω-6 varied from 7.43% to 22.95%, peaking in July. The proportion of PUFAs among total FAs ranged from 33.61% to 53.65%, peaking in September.

Table 1 shows the monthly compositions of ω-3 and ω-6. There were no significant differences in monthly ALA, GLA, and arachidonic acid (AA) over the year, but significant changes were observed in EPA and LA. Proportions of ALA, EPA, LA, GLA, and AA varied in the ranges of 15.09–27.55%, 0.09–0.16%, 7.37–15.44%, 0.72–1.61%, and 0.83–2.5%, respectively. The ratios of omega-6 to omega-3 (ω-6/ω-3) FAs varied from 0.34/1 to 0.93/1. However, there were no correlations between FA composition and environmental conditions throughout the year (Figure 9). This result indicated that environmental conditions were appropriately controlled in the ORS, resulting in sustainable biomass production and quality of *Tetraselmis* sp.

### 2.4. Annual Variation in AP and Lipid Content of Tetraselmis sp. in ORS

To further establish the feasibility of the ORS for microalgal biomass production, microalgae were cultivated in the pilot-scale ORS between November 2014 and November 2017. Several trials during the growing period of *Tetraselmis* sp. in the ORS were conducted to increase the annual average biomass productivity over the 3 years. Inter-annual AP varied from 5.64 to 44.01 g/m^2^/d (shown in Figure 10). In 2015, the average AP was not significantly different from that in 2014. In both of these years, only the water temperature was controlled (through the boiler) to avoid temperatures dropping to levels in which *Tetraselmis* sp. could not be grown. However, the major problem facing algal biomass production was contamination by predators (mostly ciliates), which led to cultivation failure and low AP in both 2014 and 2015. 

In 2016, the salinity of the culture medium was adjusted from 33 to 70 psu by the addition of NaCl to prevent contamination. This successfully prevented the appearance of predators in the ORS. As expected, average AP in 2016 was significantly higher (20.40 ± 5.81 g/m^2^/d) than those in 2014 (11.12 ± 3.28 g/m^2^/d) and 2015 (9.71 ± 2.54 g/m^2^/d). However, cultivation levels were still low. In 2017, 5% CO_2_ gas was added to the pond as a carbon source. Average AP increased in 2017 to 32.14 ± 5.56 g/m^2^/d. Moreover, maximum AP was 30.22 g/m^2^/d in 2016 and 44.01 g/m^2^/d in 2017. 

For sustainable lipid production, there should be no significant inter-annual variation in the lipid content of biomass. As shown in Figure 9, the lipid content of cells varied with PPFD. However, there were no significant differences in annual lipid content among the three years. Average lipid contents were 14.88 ± 1.26%, 14.73 ± 2.44%, 12.81 ± 2.82%, and 13.63 ± 3.42% in 2014, 2015, 2016, and 2017, respectively.

## 3. Discussion

### 3.1. Selection of Culture Medium Concentration for High Biomass Productivity

The highest biomass productivity of *Tetraselmis* sp. was observed in f medium compared to that in f/2 and f*1.5 media during the 28-day culture; therefore, f medium was selected for further cultivation experiments. In general, f/2 medium is used in growth experiments with *Tetraselmis* [36,37,38], and a higher concentration of nutrient medium has been used to attain high biomass productivity [39]. However, excessive nutrients can inhibit the growth of microalgae [40]. Therefore, culture experiments at each nutritional concentration were required to determine the appropriate concentration of medium for biomass production before large-scale cultivation. Similar to a previous report [40], PUFA (ω-3 and ω-6) composition increased with increasing nitrogen concentrations [41]. In particular, the ratio of omega-6 to omega-3 FAs is an important factor in the evaluation of PUFAs. An omega-6-to-omega-3 ratio of less than 5/1 could be used to prevent secondary cardiovascular disease, reduce cancer cell proliferation in colorectal cancer, and inhibit inflammation in rheumatoid arthritis patients [19,42]. The low ω-6/ω-3 ratios in our study under all three culture concentrations support the feasibility of mass algae production as a source of omega-3 FAs and balanced PUFA supplements.

### 3.2. Production of Biomass, Lipids, and FAs under Four Different Culture Scales

There was no significant difference in lipid content when the culture was scaled up and no correlation between culture volume and FA composition; however, the biomass productivity of *Tetraselmis* sp. decreased. Despite the well-designed culture systems, self-shading could have occurred, blocking light at a certain cell density due to the increased biomass concentration [43]. This suggests that the growth rate of *Tetraselmis* sp. decreased in the ORS due to insufficient light [44]. In the ORS, vertical mixing occurs near the paddle wheel, and laminar flow occurs throughout the culture pond [45,46], which means that cells are not exposed to light for a long time. This is why serial scale-up culture experiments are required [47,48]. In our study, the area wherein self-shading occurs was minimized by the modeling-based mixing rate despite the deep culture depth (0.4 m), and the semi-ORS was constructed in a direction that could maximize the absorption of natural light using BIM technology. For microalgae cultivation in an open raceway pond, the depth is set to range of 0.25–0.4 m, considering light penetration by self-shading [49,50]. In our previous study, we attained high biomass concentrations of 1 g/L per year (maximum 1.4 g/L) at a depth of 0.4 m [29], and several other studies have reported that high areal productivity can be achieved at a depth of 0.4 m in an open raceway pond [51,52]. As a result, there was no significant difference in biomass concentration between the 200 L culture using air-lift circulation and the 40,000 L culture using paddle-wheel circulation. As shown in Figure 4, the ratio of omega-6 to omega-3 (ω-6/ω-3, 1.37/1) FAs increased, since the composition of LA in the cells of *Tetraselmis* sp. cultured in the 200 L culture significantly increased. However, this low ratio is optimal for a PUFA source (under 5/1), and there was no significant effect of scaling up the culture volume.

### 3.3. Variations in Culture Environment and Biomass and Lipid Productivities of Tetraselmis sp. in a 1-Year ORS Study

The main disadvantages of outdoor cultivation in a seasonal climate, such as in Korea, are differences in biomass productivity and the biochemical composition of cells caused by different monthly environmental conditions [53,54,55] and a reduction in biomass concentration by the inflow of large amounts of rainfall into the microalgal pond during the rainy season [28]. Although monthly light conditions were not controlled, monthly pond temperature variations could be controlled through the boiler. As a result, there were no significant differences in AP and FA compositions and ω-6/ω-3 ratios over a 1-year cultivation using our ORS. Several studies have reported that uncontrollable light conditions cause differences in biomass productivity and FA composition in microalgae [56,57]; however, a previous study using the same strain showed no significant seasonal variations in FA composition [58], which supports our results. However, the monthly cell lipid content differed across the year, as reported in other studies [59]. Variations in lipid content could be improved through various lipid accumulation strategies that increase cell stress levels [60], such as the combined action of NaCl, Fe(III), and nitrogen deprivation in the culture medium [61], the provision of additional light using blue LEDs [62], and the application of chemical additives (EDTA) [63].

### 3.4. Variations in the AP and Lipid Content of Tetraselmis sp. in a 3-Year ORS Study

During the 3-year pilot-scale cultivation, several factors reduced biomass production. These were contamination of ciliate predators, typhoons in summer, and restricted access to the ORS pilot plant. Contamination of cultures with predators is a major problem associated with cultivation failure [64]. Especially in outdoor cultivation, microalgae can be exposed to contamination, reducing biomass concentrations and productivity within a few days [65,66,67]. Therefore, in an effort to prevent contamination, a halo-tolerant algae, *Tetraselmis* sp. [68], was used in our cultivation. The growth of predators was inhibited by increasing the salinity of the culture, resulting in a significantly higher AP. Such properties of microalgae are often used in large-scale cultivation [69], and it is known that a sudden change in salinity can damage predators without microalgal cell loss for several microalgal species [69,70]. When microalgal cells are exposed to high salinity, several processes are activated to protect the cells, such as regulating the uptake and export of sodium ions through the cell membrane, accumulation of protective solutes and stress proteins to osmotic pressure, and recovery of turgor pressure [71,72]. These processes cause intracellular stress and an increase in lipids as a reserve energy source [72,73]. However, there was no significant increase in the lipid content and no variations in FA composition (results not shown) due to increased salinity. This suggests that 70 psu did not significantly affect the growth of the *Tetraselmis* sp. strain used in this study. Therefore, this strain could be used for the sustainable production of high-value biochemical compounds annually.

Access to the pilot ORS plant was limited as it was located in a thermal power plant site. To increase annual AP despite the limited culture period available due to the restricted access, 5% CO_2_ gas was injected into the pond water. A previous study showed that biomass productivity of *Tetraselmis suecica* was increased by supplying 5% CO_2_ gas [74]. In the present study, CO_2_ gas significantly increased average annual AP but had no effect on the lipid content. Supplemental CO_2_ gas can increase biomass productivity by providing a carbon nutrient source to microalgae and by altering the pH of the culture medium [75]. However, an excessive supply of CO_2_ causes stress to microalgal cells, increasing lipid contents, decreasing biomass productivity [76], and reducing cell growth [77,78]. The injection of 5% CO_2_ was appropriate in the pilot ORS.

In 2017, semi-ORS provided 117.31 tons/ha/year biomass yield and $8.15/kg biomass production cost. Details of capital investment (30-year lifespan) and operating costs are summarized in Table 2.

Most microalgae species are known to grow in a temperature range of 15–30 °C [55]. In previous study in Korea, air temperature was positively correlated with water temperature (*p* < 0.001) [79]. Water temperature (*T_W_*, °C) could be calculated by air temperature (*T_A_*, °C) from correlation equation: *T_W_ =* 0.7781 × *T_A_* + 6.653. According to our 4-year climate data, and the above equation, the appropriate period of cultivation without a greenhouse is 6 months (May to Oct) in Korea. Assuming that the production period, the biomass yield was reduced to 58.66 tons/ha/year, and the production cost was estimated at $9.25/kg (scenario 1). However, we could cultivate for year-round by controlling the water temperature in semi-ORS. Furthermore, rainfall could be also prevented from entering the system by constructing the ORS in a greenhouse, as was done for the semi-ORS. Therefore, the biomass yield from ORS without a greenhouse is expected to decrease than an estimate, considering the inflow of rainfall (scenario 2). Despite of few references for comparing open- and semi-ORS, semi-ORS was estimated that more affordable and cost effective system than conventional open raceway pond.

## 4. Materials and Methods

### 4.1. Strain and Culture Medium

The marine green microalga *Tetraselmis* sp. MBEyh04Gc (KCTC 12432BP), which has a close relationship with *Tetraselmis striata* [80], was obtained from the Marine Bioenergy R&D Consortium (MBE) of Inha University, Incheon, Korea. The culture medium used was modified Guillard’s f/2 medium [81] (75 mg/L NaNO_3_, 5 mg/L NaH_2_PO_4_, 30 mg/L Na_2_SiO_3_.9H_2_O, 3.15 mg/L FeCl_3_·6H_2_O, 4.36 mg/L Na_2_EDTA·2H_2_O, 0.0098 mg/L CuSO_4_·5H_2_O, 0.0063 mg/L Na_2_MoO_4_·2H_2_O, 0.18 mg/L MnCl_2_·4H_2_O, 0.022 mg/L ZnSO_4_·7H_2_O, and 0.01 mg/L CoCl_2_·6H_2_O), f medium, and f*1.5 medium (which contain concentrations of nutrients double and triple those of f/2 medium, respectively) without vitamins. Each culture medium, based on natural seawater, was filtered through a GF/C filter (47 mm diameter; Whatman, Maidstone, UK) and sterilized in an autoclave (VS-1321-60, VisionScientific, Daejeon, Korea) for 2 and 5 L cultures. For 200 and 40,000 L cultures, 1% (per culture volume) of a 13% sodium hypochlorite (NaOCl) solution was administered to the culture medium at night and let stand for 24 h, and then, NaOCl was neutralized with sodium thiosulfate (1 mol/L). After that, 10% (of total volume) seed was inoculated into the pond (inoculation biomass, 0.4–0.5 g/L dry cell weight).

### 4.2. Culture Conditions

Laboratory cultures were conducted in 2 and 5 L flasks and 200 L transparent circular cylinders to investigate biomass production, lipid content, and FA profiles of *Tetraselmis* sp. Culture conditions were maintained at 27 ± 2 °C under a 12 h light:12 h dark cycle at 100 ± 5 μmol photons m^−2^s^−1^ (Li-250A, Li-COR, Lincoln, NE, USA) with a fluorescent lamp. The air flow rate was 0.4 vvm (volume air per volume culture per minute) to prevent the sedimentation of *Tetraselmis* sp. cells during the culture periods.

Additionally, 40,000 L culture was conducted with paddle-wheel circulation. Pond temperature, salinity, and pH were measured using a water quality meter (MPS556, YSI Incorporated, Chicago, IL, USA). The PPFD of the natural light source was measured with a digital photometer (Li-250A, Li-COR, Lincoln, NE, USA) at 0.2 m above the pond water surface. The pure CO_2_ gas was mixed with air through the air pump to provide a 5% concentration. The 5% CO_2_ gas was bubbled into the ponds through aquaria air stones (150 mm × 30 mm × 30 mm) with a constant flow rate of 0.1 vvm on the bottom of the pond in 2 m intervals.

### 4.3. Biomass Concentration, Productivity, and Harvesting

The biomass concentration of *Tetraselmis* sp. was estimated by sampling and filtering 20 mL of culture medium using GF/C filter paper (47 mm diameter; Whatman, Maidstone, UK). Filter paper was dried at 60 °C in a drying oven (VS-1202D3, Vision Scientific Co., Bucheon, Korea) for 24 h to obtain dried cell weights (g/L). The biomass concentration (*X*, g/L) was calculated from the difference in dry weight before and after filtration. AP (g/ m^2^/ d) is the biomass productivity per unit of ground area occupied by the culture system and was calculated using the following equation:*V* (*X*_1_ − *X*_0_)/*A* (*t*_1_ − *t*_0_)(1)
where *V* is the volume of the culture medium in L, *A* is the area of the culture system in m^2^, and *X*_1_ and *X*_0_ are the biomass concentrations collected at *t*_1_ (sampling) and time *t*_0_ (inoculation), respectively. 

*Tetraselmis* sp. biomass was harvested using a tubular separator (GQLY series, Hanil S.M.E, Anyang, Korea) at 15,750 g for 3 h. Harvested wet biomass was stored at −70 °C in a deep freezer and then lyophilized for 3 days in a freeze-dryer (FDTA-45, Operon, Gimpo, Korea).

### 4.4. Analysis of Lipid Content and FA Composition

Total lipids were determined using the Soxhlet method [82]. The crude lipid content was calculated using the following equation: *C_L_* (%) *= W_L_* (g)/*W_A_* (g) × 100(2)
where *W_L_* is the weight of the crude lipid content (%) and *W_A_* is the dry cell mass sample weight. Experiments were analyzed for EFAs such as ALA, EPA, LA, GLA, and AA. First, the microalgae cultured for 14 days were centrifuged (Sorvall RC6 plus, Thermo Fisher, Asheville, NC, USA) at 15,100× *g* for 10 min. The pellet was then lyophilized for 3 days in a freeze-dryer (FDTA-45, Operon, Gimpo, Korea). Each lyophilized cell sample was measured using FID-equipped gas chromatography (GC, Agilent 7890A, Agilent, Palo Alto, CA, USA) to analyze the FA composition of the microalgal cells. In GC analysis, the temperature was increased from 150 °C to 201 °C at 3 °C intervals for 10 min and maintained for 12 min. Then, the temperature was increased from 201 °C to 210 °C in 3 °C intervals to analyze fatty acid methyl esters. The standard used was pentadecanoic acid (C15:0, P6125, Sigma-Aldrich, MO, USA) [83].

### 4.5. System Construction and Strategies for Pilot-Scale Culture

#### 4.5.1. Culture and Pilot Production Site

Youngheung Island, located in the city of Incheon, Korea (37.1° N, 126.2° E), was selected as the site for a pilot study on the production of *Tetraselmis* sp. The average temperature from 2010 to 2014 was 12.3 ± 10.3 °C, exceeding 24 °C in summer and below −1 °C in the winter. The monthly mean temperatures are shown in Figure 11a. The number of sunny days per year ranged from 89 to 101, and the number of rainy days ranged from 95 to 117. The average annual rainfall in the cultivation site was 1379 ± 408 mm. The average weekly solar radiation ranged from 1873 to 5772 Wh/m^2^ (Figure 11b). 

#### 4.5.2. Construction of the Semi-ORS 

Figure 12 shows the construction process of the semi-ORS, which followed our previous study [29]. A pilot system was constructed near the Youngheung thermal power plant using BIM technology, which analyzes retrospective environmental data (temperature, solar radiation, wind direction, shadow effect, etc.) of the past decade or half decade and uses this detailed information to predict near future conditions to design a system with an optimal and ecofriendly structure. 

Figure 13a shows a vertical-sectional view of the ORSs based on BIM technology. The side windows were designed economically and maximized the benefit from natural ventilation. The windows were opened at >35 °C. Figure 13b shows a horizontal-sectional view of the ORSs. The size of each culture pond was 12,000 (L) × 3250 (W) × 600 (H) mm, and the culture raceway pond was installed with concrete and PVC liner under the ground to a depth of 600 mm to use geothermal heat. The culture medium was kept at a depth of 400 mm. Boiler pipes were laid in the concrete ground of ORSs to maintain the water temperature in autumn, spring, and winter.

Computational fluid dynamics was adopted to simulate and optimize the mixing characteristics of the culture medium in the ORSs. The software Flow-3D (version 10.0) was used to simulate the fluid flow to determine optimal paddle-wheel rotational speeds by ARA Consulting and Technology Company. The rotational speed selected was 15 rpm, as reported in our previous study [29].

#### 4.5.3. Maintaining the Culture System

The pond temperature was maintained between 20 and 25 °C, and the operation start temperature ranged from 21 to 23 °C in October. The boiler was operated variably based on the temperature changes (on 30 min, off 30 min), and the maximum operation was used to minimize temperature variation and energy input in winter (on 50 min and off 10 min).

### 4.6. Statistical Analysis

All statistical analyses were performed using GraphPad Prism 8.4.3 software (version 8.4.3, GraphPad Software, San Diego, CA, USA). The mean values were compared using one-way analysis of variance followed by Tukey’s test. Statistical significance was at *p* < 0.05. Pearson’s correlation coefficients were computed to determine the relationship between biochemical composition parameters (*p* < 0.05).

## 5. Conclusions

Our results demonstrated that a 240,000 L semi-ORS constructed near the northernmost area of the Republic of Korea, together with the microalga *Tetraselmis* sp. KCTC 12432BP and an appropriate concentration of culture medium, is appropriate for sustainable microalgal biomass production under Korean climate conditions. Although the biomass productivity of *Tetraselmis* sp. decreased compared to that of lab-scale cultures and the lipid content of cells varied with monthly PPFD, the productivity, FA composition, and ω-6/ω-3 ratio of the biomass cultured in the pilot-scale semi-ORS were maintained throughout the year. Furthermore, salinity was used to control contamination with predators, and injection of CO_2_ gas improved biomass productivity without affecting lipid content. 

## Figures and Tables

**Figure 1 marinedrugs-19-00314-f001:**
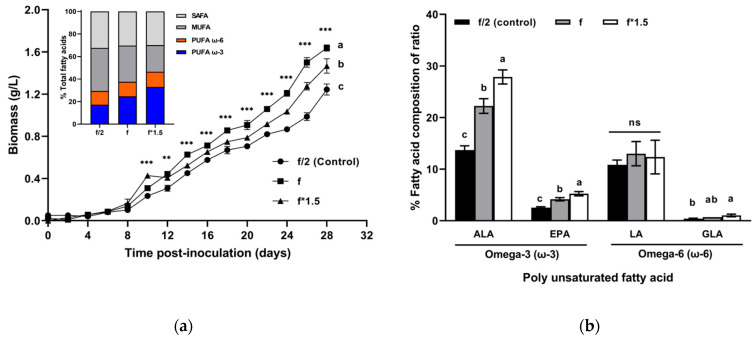
(**a**) Biomass concentration (g/L) of Tetraselmis sp. cultured under three different concentrations of media for 28 days (mean ± standard error, *n* = 3). The inset figure shows total fatty acids (%) for the compositions of saturated fatty acid (SAFA), mono-unsaturated fatty acid (MUFA), and omega-3 and omega-6 fatty acids. (**b**) Compositions of omega-3 and 6 fatty acids (%) in Tetraselmis sp. cultured under three different concentrations of media for 28 days (mean ± standard error, *n* = 3). ALA, alpha-linolenic acid; EPA, eicosapentaenoic acid; LA, linoleic acid; GLA, gamma-linolenic acid. Asterisks indicate significant differences at ** *p* < 0.002 and *** *p* < 0.001. The different lowercase letters indicate significant differences between each concentration; ‘ns’ indicates not significant (one-way analysis of variance with post-hoc Tukey’s test, *p* < 0.05).

**Figure 2 marinedrugs-19-00314-f002:**
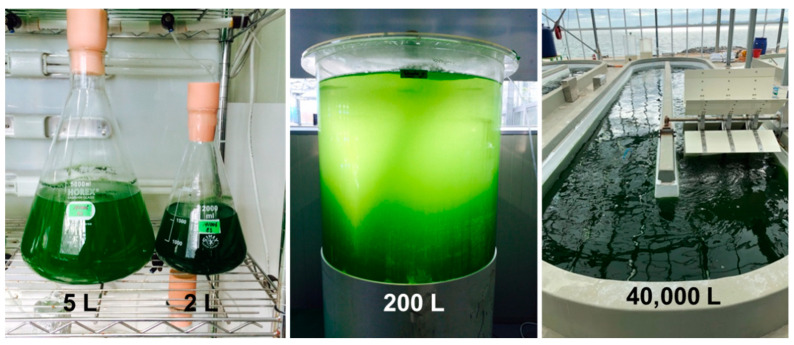
*Tetraselmis* sp. culture on 2, 5, 200, and 40,000 L scales.

**Figure 3 marinedrugs-19-00314-f003:**
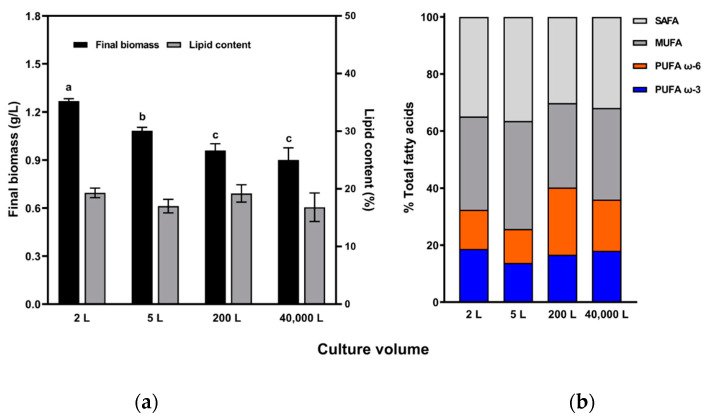
(**a**) Final biomass concentration (g/L, left Y axis) and lipid content (%, right Y axis) of *Tetraselmis* sp. cultured in four different volumes (2, 5, 200 and 40,000 L) for 28 days (mean ± standard error, *n* = 3). (**b**) Fatty acid composition (%) of *Tetraselmis* sp. cells was divided into four groups (saturated fatty acid (SAFA), mono-unsaturated fatty acid (MUFA), polyunsaturated fatty acid (PUFA) ω-6, and PUFA ω-3). Different lowercase letters indicate significant differences between each concentration (one-way analysis of variance with post-hoc Tukey’s test, *p* < 0.05).

**Figure 4 marinedrugs-19-00314-f004:**
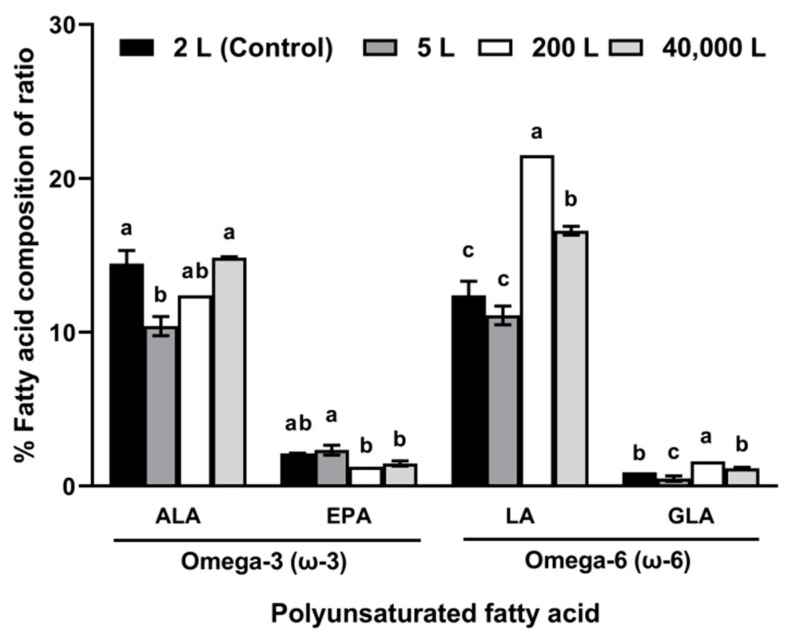
Compositions of omega-3 and 6 (%) fatty acids of *Tetraselmis* sp. cultured under four different culture scales for 28 days (mean ± standard error, *n* = 3). ALA, alpha-linolenic acid; EPA, eicosapentaenoic acid; LA, linoleic acid; GLA, gamma-linolenic acid. The different lowercase letters indicate significant differences between each scale (one-way analysis of variance with post-hoc Tukey’s test, *p* < 0.05).

**Figure 5 marinedrugs-19-00314-f005:**
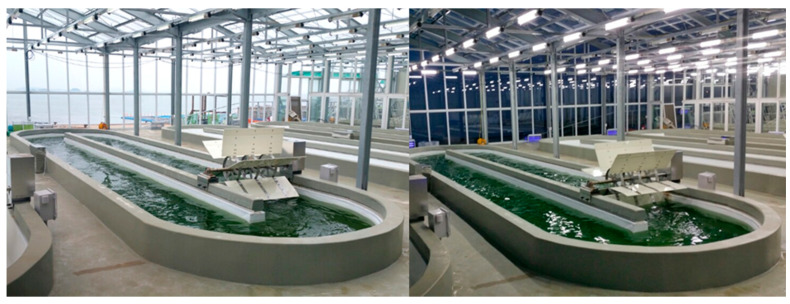
Photographs of the facility used for year-round cultivation of *Tetraselmis* sp. in six 40,000 L semi-open raceway systems (ORSs; each pond size, 12,000 (L) × 3250 (W) × 400 (D) mm; rotational speed, 15 rpm).

**Figure 6 marinedrugs-19-00314-f006:**
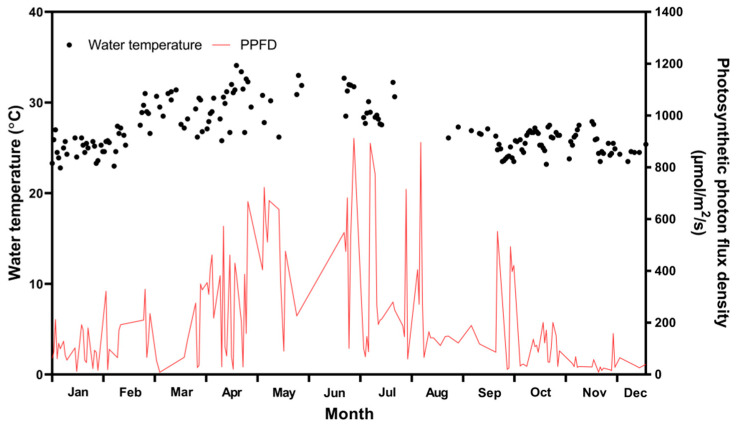
Pond water (medium) temperature (left Y axis) and photosynthetic photon flux density (PPFD; right Y axis) in *Tetraselmis* sp. culture ponds from 1 January to 31 December.

**Figure 7 marinedrugs-19-00314-f007:**
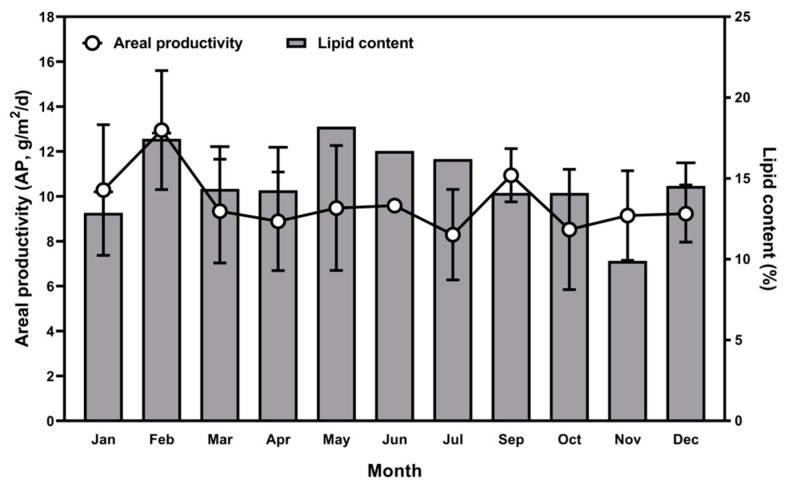
Monthly average areal productivity (left Y axis) and lipid content (right Y axis) of *Tetraselmis* sp. from 1 January to 31 December.

**Figure 8 marinedrugs-19-00314-f008:**
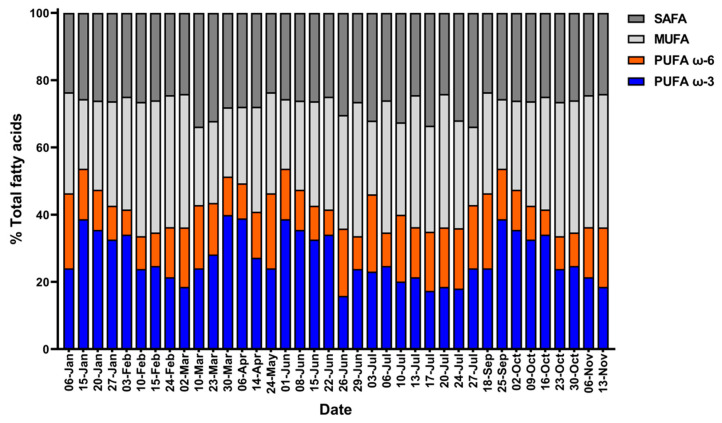
Daily fatty acid composition of *Tetraselmis* sp. cells divided into four groups (saturated fatty acid (SAFA), mono-unsaturated fatty acid (MUFA), polyunsaturated fatty acid (PUFA) ω-6, and PUFA ω-3).

**Figure 9 marinedrugs-19-00314-f009:**
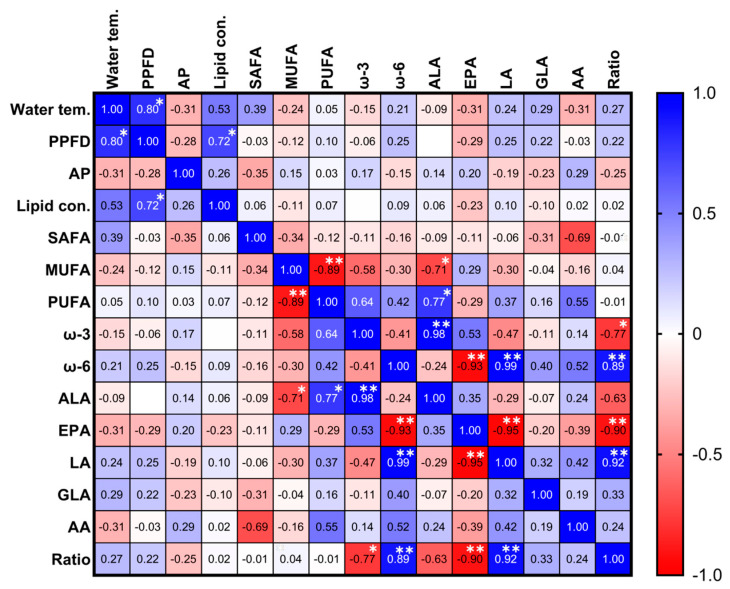
Pearson’s correlation coefficient (r) matrix of environmental conditions (water temperature and PPFD), AP, lipid content, FA composition (SAFA, MUFA, PUFA, ω-3, ω-6, ALA, EPA, LA, GLA, and AA), and ratio. Water tem., water temperature; PPFD, photosynthetic photon flux density; AP, areal productivity; SAFA, saturated fatty acid; MUFA, mono-unsaturated fatty acid; PUFA, polyunsaturated fatty acid; ω-3, omega-3; ω-6, omega-6; ALA, alpha-linolenic acid; EPA, eicosapentaenoic acid; LA, linoleic acid; GLA, gamma-linolenic acid; AA, arachidonic acid; Ratio, omega-6/omega-3. Asterisks indicate significant differences at * *p* < 0.05 and ** *p* < 0.002.

**Figure 10 marinedrugs-19-00314-f010:**
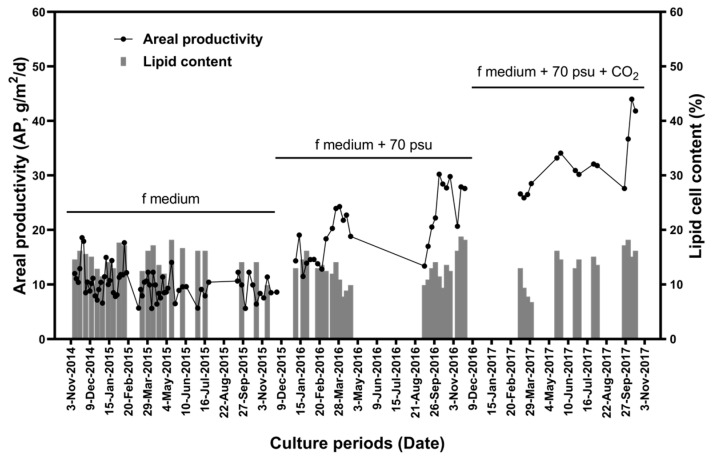
Areal productivity and lipid content of *Tetraselmis* sp. cells from November 2014 to October 2017.

**Figure 11 marinedrugs-19-00314-f011:**
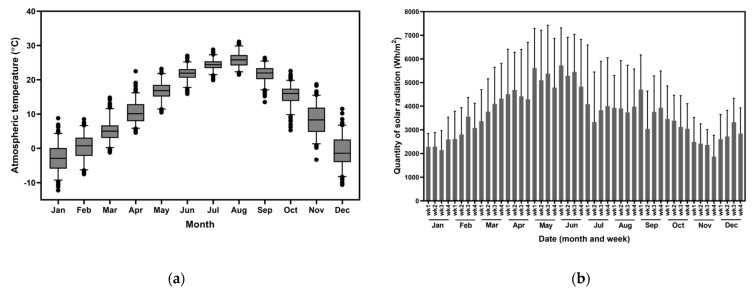
Environmental conditions of the study site. (**a**) Mean monthly atmospheric temperature (°C); (**b**) integrated weekly solar radiation (Wh/m^2^) for 2010–2014.

**Figure 12 marinedrugs-19-00314-f012:**
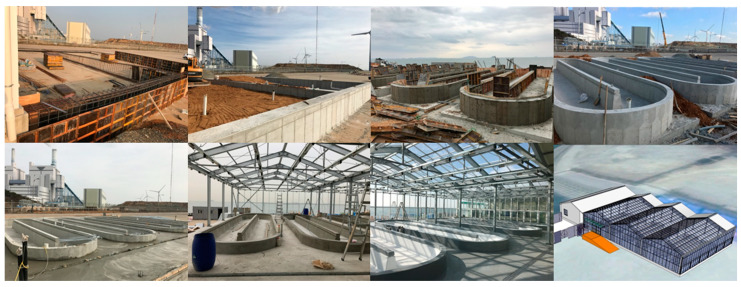
Images showing the construction of the semi-open raceway system (ORS) at the selected site in Incheon, Korea.

**Figure 13 marinedrugs-19-00314-f013:**
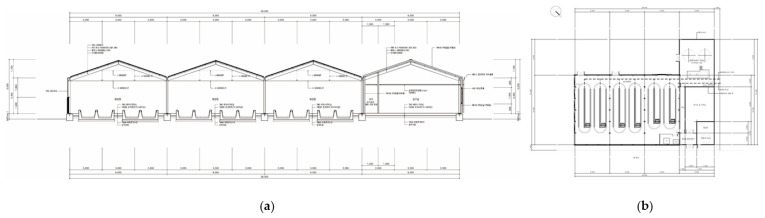
A cross-sectional view of the pilot microalgae production system using building information modeling (BIM) analysis (patent no. 10-1142,358, 10-1110,068). (**a**) Vertical view of the system; (**b**) horizontal view of the system.

**Table 1 marinedrugs-19-00314-t001:** Monthly variation in omega-3 and omega-6 (%, per total fatty acids) in *Tetraselmis* sp. cells cultured in a semi-open raceway system for 1 year.

Month	ALA (C18:3n3)	EPA (C20:5n3)	LA (C18:2n6)	GLA (C18:3n6)	AA (C20:4n6)	Ratio (ω-6/ω-3)
Jan	27.55 ± 5.00	0.12 ± 0.04 ^ab^	11.86 ± 4.69 ^ab^	0.88 ± 0.36	1.91 ± 0.40	0.49/1
Feb	20.21 ± 4.90	0.16 ± 0.02 ^a^	8.02 ± 2.89 ^ab^	0.85 ± 0.27	1.50 ± 0.11	0.43/1
Mar	21.56 ± 8.64	0.07 ± 0.02 ^b^	13.38 ± 2.53 ^ab^	0.72 ± 0.42	1.43 ± 0.37	0.70/1
Apr	27.33 ± 7.75	0.11 ± 0.01 ^ab^	9.01 ± 1.08 ^ab^	1.61 ± 1.18	1.26 ± 0.05	0.39/1
May	21.13	0.08	18.21	1.41	2.5	0.93/1
Jun	24.61 ± 7.48	0.15 ± 0.03 ^a^	10.00 ± 4.71 ^ab^	0.83 ± 0.36	1.36 ± 0.68	0.49/1
Jul	16.88 ± 2.55	0.09 ± 0.04 ^b^	15.44 ± 4.27 ^a^	1.04 ± 0.29	0.83 ± 0.90	0.85/1
Sep	26.94 ± 8.22	0.09 ± 0.02 ^ab^	15.34 ± 4.05 ^ab^	1.05 ± 0.51	2.07 ± 0.61	0.66/1
Oct	24.19 ± 5.19	0.16 ± 0.01 ^a^	7.37 ± 1.46 ^b^	0.72 ± 0.12	1.59 ± 0.17	0.34/1
Nov	15.09 ± 1.60	0.12 ± 0.02 ^ab^	13.16 ± 1.61 ^ab^	1.32 ± 0.16	1.68 ± 0.21	0.83/1

ALA, alpha-linolenic acid; EPA, eicosapentaenoic acid; LA, linoleic acid; GLA, gamma-linolenic acid; AA, arachidonic acid. Different lowercase letters next to numbers indicate significant differences between months (one-way analysis of variance with post-hoc Tukey’s test, *p* < 0.05).

**Table 2 marinedrugs-19-00314-t002:** Biomass production costs of *Tetraselmis* sp. in semi-open raceway system.

	Semi-ORS (a)	Scenario 1 (b)	Scenario 2 (c)
Scale (ha)	0.04	0.04	0.04
Biomass yield			
g/m^2^/d	32.14	32.14	<32.14
tons/ha/year	117.31	58.66	<58.66
Capital cost ($)			
Major purchased equipment	25,000	25,000	25,000
Installation	70,000	70,000	70,000
Construction	390,000	-	-
Infrastructure	141,000	141,000	141,000
Other	4500	4500	4500
Total capital costs	630,500	240,500	240,500
Depreciation (30 years, $/year)	24,017	11,017	11,017
Operating cost ($/year)			
Fertilizers	3000	1500	1500
Labor	5000	5000	5000
Electricity	4500	2250	2250
Water	-	-	-
Analytical cost	2200	2200	2200
Total operating costs	14,700	10,950	10,950
Total production cost ($/year)	38,717	21,967	21,967
Biomass production cost ($/kg)	8.15	9.25	>9.25

(a) Semi-open raceway system with a greenhouse used in this study; (b) Open raceway system without a greenhouse (6 months period); (c) Open raceway system without a greenhouse (6 months period, rainfall inflow); Land cost of semi-ORS ($3000/year) is included in depreciation; Depreciation periods were assumed to be 30 years.

## Data Availability

The data presented in this study are available in insert article.

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
