# Peer review of "Year-Round Cultivation of Tetraselmis sp. for Essential Lipid Production in a Semi-Open Raceway System"

_marinedrugs, 2021, doi:10.3390/md19060314_

Round 1
Reviewer 1 Report
The manuscript marinedrugs-1220529 entitled " Year-round cultivation of Tetraselmis sp. for bioactive lipid production in a semi-open raceway system" describes a 3-year trial using raceways inside a greenhouse. The data presented is interesting and can even be publishable, but my main concern about it is connected to the fact that the authors claims about the feasibility of the whole project is not substantiated with hard data concerning capital and operational costs using the proposed semi-ORS. Therefore, the authors need to:
- Include estimates of the costs of the biomass grown with and without the greenhouse to protect the cultures during the cold months in Korea (please include the delta of the additional costs per Kg of biomass produced with and without the greenhouse)
- What would be the operational window if no greenhouse is constructed
- Discuss also the volumetric productivity (the authors only mention the areal productivity) and the impact of using such high water column heights (0.4 m); in general, in raceways, column heights higher than 0.02 m are rarely used because that decreases dramatically the productivity of the whole system; compare your productivity with those of other systems published so far. The authors need to improve their discussion significantly in order to better understand the added value of the proposed system.
Without this thorough discussion, the manuscript has little value, because it will be hard to compare with productivity data published previously. Moreover, its true feasibility is also hard to assess if the production costs of building the raceways AND the greenhouses are not discussed.
Another thing I would like for the authors to improve is the classification of their strain: Please use molecular genetics and indicate exactly to which clade their Tetraselmis sp. strain belongs to. This does not seem too hard to be done.
Other comments are given in the manuscript itself attached.

Author Response
Response to the reviewers’ comments
Manuscript ID: marinedrugs-1220529
Title: Year-round cultivation of Tetraselmis sp. for bioactive lipid production in a semi-open raceway system
Dear Editor:
We are very thankful to the editor-in-chief and reviewers for their valuable effort in reviewing our manuscript and providing suggestions to improve our manuscript. We have taken all of the comments and suggestions pointed out by the reviewers into consideration with this revision. Thus, we believe that our manuscript can now meet the journal’s quality criteria for publication. Our point-by-point responses are provided below. We hope that we have provided appropriate answers to all concerns. The changes are indicated by red in the revised manuscript.
Thank you for your consideration. I look forward to hearing from you.
Sincerely,
Do-Hyung Kang
Jeju Marine Research Center
Korea Institute of Ocean Science and Technology (KIOST)
Department of Ocean Science (Oceanography), KIOST School
University of Science and Technology (UST)
E-mail: dohkang@kiost.ac.kr
Tel: 82-64-798-6010
Reviewers' comments:
Reviewer #1 :
Reviewer comment 01 :
The manuscript marinedrugs-1220529 entitled " Year-round cultivation of Tetraselmis sp. for bioactive lipid production in a semi-open raceway system" describes a 3-year trial using raceways inside a greenhouse. The data presented is interesting and can even be publishable, but my main concern about it is connected to the fact that the authors claims about the feasibility of the whole project is not substantiated with hard data concerning capital and operational costs using the proposed semi-ORS. Therefore, the authors need to:
Author’s response :
Based on 2017 cultivation using semi-ORSs, costs of biomass production were estimated at $8.15/kg and, as you suggested, the details of captical investment and operating cost were added in "Discussion" section as Table 2 (lines 364-369, pages 11-12 in revised MS).
Reviewer comment 02 :
Include estimates of the costs of the biomass grown with and without the greenhouse to protect the cultures during the cold months in Korea (please include the delta of the additional costs per Kg of biomass produced with and without the greenhouse)
Author’s response :
We have not performed annual cultivation in an open raceway pond without a greenhouse. However, in a previous study, when a water temperature control system was installed in an open raceway pond in Korea, enhanced (44% per year) microalgal biomass productivity (132% in winter) was observed, and net profit ($67160/year) was enhanced by 95% when compared to that of the typical open raceway pond ($34380/year) to produce 1,000 tons of biomass annually. As you suggested, the relevant sentences were added to the ‘Discussion’ section (lines 315-319, page 10-11 in revised MS).
Reviewer comment 03 :
What would be the operational window if no greenhouse is constructed?
Author’s response :
Most microalgae are known to grow at 15–30 °C, and according to our 4-year climate data, 46% of the year (May to Oct) in Korea is > 15 °C. As you suggested, the relevant sentences were added to the ‘Discussion’ section (lines 313-315, page 10 in revised MS).
Reviewer comment 04 :
Discuss also the volumetric productivity (the authors only mention the areal productivity) and the impact of using such high water column heights (0.4 m); in general, in raceways, column heights higher than 0.02 m are rarely used because that decreases dramatically the productivity of the whole system; compare your productivity with those of other systems published so far. The authors need to improve their discussion significantly in order to better understand the added value of the proposed system. Without this thorough discussion, the manuscript has little value, because it will be hard to compare with productivity data published previously. Moreover, its true feasibility is also hard to assess if the production costs of building the raceways AND the greenhouses are not discussed.
Author’s response :
Is the depth of 0.2 m? For microalgae cultivation in an open raceway pond, the depth is set to a range of 0.25–0.4 m, considering light penetration by self-shading. In our previous study, we attained high biomass concentrations of 1 g/L per year (maximum 1.4 g/L) with a depth of 0.4 m, and several other studies have reported that high areal productivity can be achieved with a depth of 0.4 m in an open raceway pond. Moreover, our aim was to cultivate during four seasons, and to increase areal productivity due to expensive land cost of Korea. To explain these reasons, as you suggested, the relevant sentences were added to the ‘Discussion’ section (lines 295-300, page 10 in revised MS).
Reviewer comment 05 :
Another thing I would like for the authors to improve is the classification of their strain: Please use molecular genetics and indicate exactly to which clade their Tetraselmis sp. strain belongs to. This does not seem too hard to be done.
Author’s response :
The strain used for cultivation was Tetraselmis sp. MBEyh04Gc (KCTC 12432BP) provided by Inha University. It was reported to be closely related to Tetraselmis striata. As you suggested, detail information of species with reference was added to the ‘Materials and Methods’ section (lines 372-373, page 12 in revised MS).
Other comments are given in the manuscript itself attached.
Reviewer comment 06 :
Line22: “Culture volume of 40 ton”. Volumes are not measured in tons; please revise
Author’s response :
As you suggested, all “40 ton” in manuscript was edited to “40,000 L”.
Reviewer comment 07 :
Line57: “on”. for
Author’s response :
As you suggested, “on” was edited to “for” (line 59, page 2 in revised MS).
Reviewer comment 08 :
Line65-66: “However, they require high operating costs, initial investments, device maintenance, and it is difficult to expand facilities”. This sentence should be improved. What intial investments? The sentence is also gramatically incorrect.
Author’s response :
As you suggested, the above sentence was edited to “However, they require high operating costs, capital investment costs, device maintenance, and it can be difficult to expand facilities” (lines 67-68, page 2 in revised MS).
Reviewer comment 09 :
Line76: “quality of biochemical compositions”. Vague and awkwardly formulated. Please improve this.
Line82: “quality of biochemical composition”. See previous comment.
Author’s response :
As you suggested, “quality of biochemical composition” was awkwardly formulated, so edited to “biochemical compositions” (lines 78 and 83, page 2 in revised MS).
Reviewer comment 10 :
Line83: “semi-ORSs”. What are semi-ORS? Please define this.
Author’s response :
As you suggested to explaine the semi-ORS, “In 2011, we established ORSs in which open raceway ponds were installed in a greenhouse (so called semi-ORSs)” was added in “Introduction” section (lines 84-85, page 2 in revised MS).
Reviewer comment 11 :
Line89: “considered beneficial”. Replace with: “considered to be beneficial”.
Author’s response :
As you suggested, “considered beneficial” was edited to “considered to be beneficial” (line 92, page 2 in revised MS).
Reviewer comment 12 :
Line106: “17.33 ± 0.98 (f/2 medium) to 24.72 ± 1.80 and 33.10 ± 1.59% in”. % of what?
Author’s response :
As you suggested to clarify the “%”, “Compositions of fatty acids (%)” was edited to “Compositions of FAs (%, per total FAs)” (line 107, page 3 in revised MS).
Reviewer comment 13 :
Line 109: “Omega-3 FAs increased in a concentration-dependent manner”. It is important to state whether is the % or the absolute concentration of PUFA that changed. It is difficult to asceratain that, as authors do no show the lipid levels of each culture.
Author’s response :
As you suggested, the above sentence was edited to “The proportion of omega-3 among total FAs increased in a concentration-dependent manner” (lines 111-112, page 3 in revised MS).
Reviewer comment 14 :
Line 113: “Among the omega-6 compositions”. Awkwardly formulated. Please revise.
Author’s response :
As you suggested, the above sentence was edited to “Regarding omega-6 compositions” (line 116, page 3 in revised MS).
Reviewer comment 15 :
Line115: “differentmedia”. Please separate the words.
Author’s response :
As you suggested, “differentmedia” was edited to “different medium” (line 118, page 3 in revised MS).
Reviewer comment 16 :
Line123-128: “Figure 1.(a) Biomass concetration …indicate significant”. What does ns mean? Not significant? Please add this to the legend.
Author’s response :
As you suggested, “; ‘ns’ indicates not significant” was added in ‘Figure 1’ legends (line 132, page 3 in revised MS).
Reviewer comment 17 :
Line156-157: “Compare with the control (2-L culture) in the 40-ton culture, the composition of LA significantly increased (14.85 ± 0.07%),”. Awkwardly formulated. Please revise.
Author’s response :
As you suggested to clarify, above sentence was edited to “Compared with that in the control (2 L culture), the composition of LA significantly increased (14.85 ± 0.07%) in the 40,000 L culture,” (lines 160-161, page 5 in revised MS).
Reviewer comment 18 :
Line179: “controlled”.This is somewhat strange and even misleading, because temperature nor PPF were controlled. If they were, that is not clear now.
Author’s response :
Temperature could be controlled, but not PPFD in 40,000 L scale culture. So as you suggested, I clarified the culture conditions to “The average pond temperature was controlled at 27.35 ± 2.76 °C, and PPFD, which could not be controlled, was 189.21 ± 200.14 µmol photon m-2s-1 ” (line 183-185, page 5 in revised MS).
Reviewer comment 19 :
Line196: “Lipid content was positively correlated with PPFD over the year (r = 0.72)”. r-values by themselves are meaningless unless you include the corresponding p-value.
Author’s response :
As you suggested, r-value was edited to p-value (p = 0.019) (line 202-203, page 6 in revised MS).
"Please see the attachment"
Reviewer 2 Report
Review comments on “Year-round cultivation of Tetraselmis sp. for bioactive lipid production in a semi-open raceway system”
General comment:
As the manuscript only dealt with the identification of essential fatty acids and not even any in vitro bioactive assays, I think the present title may slightly be changed as “Year-round cultivation of Tetraselmis sp. for essential lipid production in a semi-open raceway system”. However, in the discussion as a possibility for bioactive lipid production, the following article may be referred to.
https://www.mdpi.com/1660-3397/19/1/28
“Ton” is the unit of mass, so avoid using this term for liquid volume, change appropriately, possibly as “L” (litre).
Write always the genus name in italics as “Tetraselmis sp.” throughout the manuscript.
Check for typographical errors below:
Line98: “2.1. Selection of culture medium concentration for high biomass and fatty acid prodcutivity"
Line259: “3.1. Selection of culture medium concentration for high biomass prodcutivity"
Line294: “3.3. Varioations in culture environment and biomass and lipid productivities…”
Specific comments:
Line 185: “Figure 5. Photographs of the facility used for year-round cultivation of Tetraselmis sp. in six 40-ton semi-ORSs” Provide details of the raceways ponds size (Length, breadth, and depth) also mention the speed of the paddle wheel.
Lines210-213: “There were no significant differences in monthly ALA, GLA, and AA over the year, but slight changes were observed in EPA and LA. As a proportion of FA, ALA, EPA, LA, GLA, and AA varied from 15.09% to 27.55%, from 0.09% to 0.16%, from 7.37% to 15.44%, from 0.72% to 1.61%, and from 0.83% to 2.5%, respectively.”
Re-write this result part of the table, which is very confusing in reading. Please use simple and small sentences.
Line220: “Table 1. Monthly variation in omega-3 and 6 (%) of Tetraselmis sp. cells cultured in a semi-open raceway system for one year”. Write “omega-3 and omega-6”, also mention clearly if this “%” per dry weight biomass?
Line359: “4.2. Culture conditions”: Mention the type of light source for 2-5L, 200L culture as well as ORS cultures. Are they all received the similar type of lights eg. LED or Fluorescent, etc. Detail how 5% CO2 was mixed/prepared, and how it was introduced in large raceways? Any measurements of amount of total CO2 supplied per day?
Line357-358:” for small volumes or with 1% (of total volume) of sodium hypochlorite for volumes above 200 L.” What about for the 200L volume? Mention how you sterilised 200L volume culture. Write the appropriate method of sterilisation for ORS system, for example, how long and at what concentration hypochlorite treatment was provided and then, how the medium was neutralised? When the culture inoculum was introduced, and from at what culture volume? Any measurements?
Author Response
Response to the reviewers’ comments
Manuscript ID: marinedrugs-1220529
Title: Year-round cultivation of Tetraselmis sp. for bioactive lipid production in a semi-open raceway system
Dear Editor:
We are very thankful to the editor-in-chief and reviewers for their valuable effort in reviewing our manuscript and providing suggestions to improve our manuscript. We have taken all of the comments and suggestions pointed out by the reviewers into consideration with this revision. Thus, we believe that our manuscript can now meet the journal’s quality criteria for publication. Our point-by-point responses are provided below. We hope that we have provided appropriate answers to all concerns. The changes are indicated by red in the revised manuscript.
Thank you for your consideration. I look forward to hearing from you.
Sincerely,
Do-Hyung Kang
Jeju Marine Research Center
Korea Institute of Ocean Science and Technology (KIOST)
Department of Ocean Science (Oceanography), KIOST School
University of Science and Technology (UST)
E-mail: dohkang@kiost.ac.kr
Tel: 82-64-798-6010
Reviewers' comments:
Reviewer #2 :
As the manuscript only dealt with the identification of essential fatty acids and not even any in vitro bioactive assays, I think the present title may slightly be changed as “Year-round cultivation of Tetraselmis sp. for essential lipid production in a semi-open raceway system”. However, in the discussion as a possibility for bioactive lipid production, the following article may be referred to.
https://www.mdpi.com/1660-3397/19/1/28
Author’s response :
As you recommended, the title was changed to “Year-round cultivation of Tetraselmis sp. for essential lipid production in a semi-open raceway system”, and the phrase ‘bioactive lipids from marine microalga’ was added to provide a better and more detailed explanation, in addition to providing a reference as follows (lines 43-44, page 1 in revised MS):
Tetraselmis (Chlorophyta), which has previously been reported to contain a substantial lipid fraction, is a potentially promising bioactive feedstock, like other microalgal species [9,10].
Reference:
- Shiels, K.; Tsoupras, A.; Lordan, R.; Nasopoulou, C.; Zabetakis, I.; Murray, P.; Saha, S.K. Bioactive Lipids of Marine Microalga Chlorococcum sp. SABC 012504 with Anti-Inflammatory and Anti-Thrombotic Activities. Marine Drugs 2021, 19.
Reviewer comment 01 :
“Ton” is the unit of mass, so avoid using this term for liquid volume, change appropriately, possibly as “L” (Litre).
Author’s response :
As you suggested, all references to “Ton” included in the manuscript have been corrected to “L” (liter) units.
Reviewer comment 02 :
Write always the genus name in italics as “Tetraselmis sp.” throughout the manuscript.
Author’s response :
As you suggested, all instances of “Tetraselmis sp.” have been corrected to “Tetraselmis sp.” throughout the manuscript.
Reviewer comment 03 :
Check for typographical errors below:
Line98: “2.1. Selection of culture medium concentration for high biomass and fatty acid prodcutivity”
Line 259: “3.1. Selection of culture medium concentration for high biomass prodcutivity”
Line 294: “3.3. Varioations in culture environment and biomass and lipid productivities…”
Author’s response :
As you suggested, the above sentences were corrected to “2.1. Selection of culture medium concentration for high biomass and FA productivity” (line 100, page 3 in revised MS), “3.1. Selection of culture medium concentration for high biomass productivity” (line 267, page 9 in revised MS), and “3.3. Variations in culture environment and biomass and lipid productivities…” (line 307, page 10 in revised MS), respectively.
Reviewer comment 04 :
Line 185: “Figure 5. Photographs of the facility used for year-round cultivation of Tetraselmis sp. in six 40-ton semi-ORSs” Provide details of the raceways ponds size (Length, breadth, and depth) also mention the speed of the paddle wheel.
Author’s response :
As you recommended, the above sentence was edited to “Figure 5. Photographs of the facility used for year-round cultivation of Tetraselmis sp. in six 40,000 L semi-open raceway systems (ORSs; each pond size, 12,000 (L) × 3250 (W) × 400 (D) mm; rotational speed, 15 rpm) (lines 190-192, page 6 in revised MS).
Reviewer comment 05 :
Lines210-213: “There were no significant differences in monthly ALA, GLA, and AA over the year, but slight changes were observed in EPA and LA. As a proportion of FA, ALA, EPA, LA, GLA, and AA varied from 15.09% to 27.55%, from 0.09% to 0.16%, from 7.37% to 15.44%, from 0.72% to 1.61%, and from 0.83% to 2.5%, respectively.” Re-write this result part of the table, which is very confusing in reading. Please use simple and small sentences.
Author’s response :
To clarify the description provided, these sentences were changed to “There were no significant differences in monthly ALA, GLA, and arachidonic acid (AA) over the year, but significant changes were observed in EPA and LA. Proportions of ALA, EPA, LA, GLA, and AA varied in the ranges of 15.09–27.55%, 0.09–0.16%, 7.37–15.44%, 0.72–1.61%, and 0.83–2.5%, respectively.” (lines 217-221, page 7 in revised MS).
Reviewer comment 06 :
Line220: “Table 1. Monthly variation in omega-3 and 6 (%) of Tetraselmis sp. cells cultured in a semi-open raceway system for one year”. Write “omega-3 and omega-6”, also mention clearly if this “%” per dry weight biomass?
Author’s response :
As you suggested, this was edited to “Table 1. Monthly variation in omega-3 and omega-6 (%, per total fatty acids) in Tetraselmis sp. cells cultured in a semi-open raceway system for 1 year” (lines 227-228, page 8 in revised MS).
Reviewer comment 07 :
Line359: “4.2. Culture conditions”: Mention the type of light source for 2-5L, 200L culture as well as ORS cultures. Are they all received the similar type of lights eg. LED or Fluorescent, etc. Detail how 5% CO2 was mixed/prepared, and how it was introduced in large raceways? Any measurements of amount of total CO2 supplied per day?
Author’s response :
Type of light soure was from fluorescent lamp in 2, 5 and 200 L scale cultures, and from natural light soure in 40,000 L scale cultivation (lines 391-392, 396, page 12 in revised MS). The CO2 gas was mixed with air thouth the air pump to provide a 5% concentration, and bubbled into the ponds with 0.1 vvm flow rate. As you suggested, the details of light soures and way to supply CO2 gas were added to ‘Materials and Methods’ section (lines 398-401, page 12 in revised MS).
Reviewer comment 08 :
Line357-358:” for small volumes or with 1% (of total volume) of sodium hypochlorite for volumes above 200 L.” What about for the 200L volume? Mention how you sterilised 200L volume culture. Write the appropriate method of sterilisation for ORS system, for example, how long and at what concentration hypochlorite treatment was provided and then, how the medium was neutralised? When the culture inoculum was introduced, and from at what culture volume? Any measurements?
Author’s response :
Autoclave process was conducted in 2 and 5 L scale cultures. In 200 and 40,000 L, The 1% (per culture volume) of 13% sodium hypochlorite solution was treated to pond for sterilization. And then culture medium was neutralizated by 1 mol/L sodium thiosulfate. 10% (of total volume) microalgal seed was inoculated after neutralization process, which was 0.4 to 0.5 g/L dry cell weight. As you suggested, details about sterlization, neutralization and inoculation processes were added in ‘Materials and Methods’ section (lines 382-386, page 12 in revised MS).
Round 2
Reviewer 1 Report
Most of my comments were answered and accommodated. However, a meaningful comparison between costs / productivities with and without the construction of the greenhouses has not been done. The authors are highly encouraged to provide such a discussion. The whole point of the manuscript is that this method is more advantageous than conventional raceway ponds. With these hard data, the main point of the whole manuscript falls apart. Please discuss the data of Table 2 and provide the biomass costs with and without the greenhouses. Use the data provided in page 10 to compare the two scenarios. Proof your point with numbers. Without them, I will not be convinced.
Author Response
Reviewers' second comments:
Reviewer #1 :
Reviewer comment 01 :
Most of my comments were answered and accommodated. However, a meaningful comparison between costs / productivities with and without the construction of the greenhouses has not been done. The authors are highly encouraged to provide such a discussion. The whole point of the manuscript is that this method is more advantageous than conventional raceway ponds. With these hard data, the main point of the whole manuscript falls apart. Please discuss the data of Table 2 and provide the biomass costs with and without the greenhouses. Use the data provided in page 10 to compare the two scenarios. Proof your point with numbers. Without them, I will not be convinced.
Author’s response :
Compared to conventional open raceway pond (2 g/m2/d; $16.51/kg), biomass productivity was higher and production cost was lower in semi-ORS (32.14 g/m2/d; $8.15/kg). As you suggested, biomass productivity and production costs of comparisons between with and without a greenhouse were added to ‘Discussion’ section with Table 2 (lines 357-378, pages 11-12 in revised MS).

Round 3
Reviewer 1 Report
The comparison done by the authors for Table 2 is an unfair one. They are comparing a fast-growing species (Tetraselmis) with a slow-grower (Dunaliella). Moreover, the scales given are totally different. Therefore, in my opinion, the manuscript does not prove its main point: the advantage of using semi-ORS. Once again, I repeat the authors should give an estimate of using the same species with or without the greenhouses. If they want they can consider the downtime during the winter as a way to demonstrate that the semi-ORS could generate a larger revenue than the open raceway ponds that would offset the higher costs of using greenhouses. However, the comparison made by the authors is invalid, because they are comparing two different species with different growth capabilities and therefore productivities.
Author Response
Reviewers' third comments:
Reviewer #1 :
Reviewer comment 01 :
The comparison done by the authors for Table 2 is an unfair one. They are comparing a fast-growing species (Tetraselmis) with a slow-grower (Dunaliella). Moreover, the scales given are totally different. Therefore, in my opinion, the manuscript does not prove its main point: the advantage of using semi-ORS. Once again, I repeat the authors should give an estimate of using the same species with or without the greenhouses. If they want they can consider the downtime during the winter as a way to demonstrate that the semi-ORS could generate a larger revenue than the open raceway ponds that would offset the higher costs of using greenhouses. However, the comparison made by the authors is invalid, because they are comparing two different species with different growth capabilities and therefore productivities.
Author’s response :
In this study, the biomass yield in semi-ORS attained 117.31 tons/ha/year, and cost of biomass production was estimated at $8.15/kg. To prove the feasibility of semi-ORS with greenhouse, we compared biomass yield and production cost as with and without a greenhouse (58.66 tons/ha/year; $9.25/kg) in Table 2. Based on our 4-year climate data and previous study in Korea, we estimated the appropriate culture period. Furthermore, considering rainfall inflow, biomass yield is expected to be decreased. As you suggested, the details of comparisons were added in ‘Discussion’ section (lines 357-381, pages 11-12 in revised MS).
